# Pharmacokinetic Evaluation of Oral Viscous Budesonide in Paediatric Patients with Eosinophilic Oesophagitis in Repaired Oesophageal Atresia

**DOI:** 10.3390/pharmaceutics16070872

**Published:** 2024-06-28

**Authors:** Raffaele Simeoli, Sebastiano A. G. Lava, Alessandro Di Deo, Marco Roversi, Sara Cairoli, Renato Tambucci, Francesca Rea, Monica Malamisura, Giulia Angelino, Isabella Biondi, Alessandra Simonetti, Paola De Angelis, Carlo Dionisi Vici, Paolo Rossi, Giuseppe Pontrelli, Oscar Della Pasqua, Bianca Maria Goffredo

**Affiliations:** 1Division of Metabolic Diseases, Bambino Gesù Children’s Hospital, IRCCS, 00165 Rome, Italy; raffaele.simeoli@opbg.net (R.S.); sara.cairoli@opbg.net (S.C.); carlo.dionisivici@opbg.net (C.D.V.); 2Clinical Pharmacology & Therapeutics Group, University College London, London WC1N 3JH, UK; s.lava@ucl.ac.uk (S.A.G.L.); alessandro.deo.19@ucl.ac.uk (A.D.D.); o.dellapasqua@ucl.ac.uk (O.D.P.); 3PhD Program in Immunology, Molecular Medicine and Applied Biotechnology, University of Rome Tor Vergata, 00133 Rome, Italy; marco.roversi@opbg.net; 4Digestive Endoscopy Unit, Bambino Gesù Children’s Hospital, IRCCS, 00165 Rome, Italy; renato.tambucci@opbg.net (R.T.); francesca.rea@opbg.net (F.R.); monica.malamisura@opbg.net (M.M.); giulia.angelino@opbg.net (G.A.); paola.deangelis@opbg.net (P.D.A.); 5Centre of Excellence for the Development and Implementation of Medicines, Vaccines, and Medical Devices for Pediatric Use, Bambino Gesù Children’s Hospital, IRCCS, 00165 Rome, Italy; isabella.biondi@opbg.net (I.B.); alessandra.simonetti@opbg.net (A.S.); paolo.rossi@opbg.net (P.R.); giuseppe.pontrelli@opbg.net (G.P.)

**Keywords:** eosinophilic oesophagitis (EoE), oesophageal atresia (EA), pharmacokinetics, oral viscous budesonide, population pharmacokinetic modelling, systemic absorption, LC-MS/MS

## Abstract

Eosinophilic oesophagitis is a long-term complication of oesophageal atresia (EA), an uncommon condition that affects approximately 1 in 3500 infants. An exploratory, open-label phase 2 clinical trial was conducted in paediatric eosinophilic oesophagitis after oesophageal atresia (EoE-EA) to assess the safety, pharmacokinetics, and efficacy of oral viscous budesonide (OVB). In total, eight patients were enrolled in the study and assigned to a twice-daily dosing regimen of either 0.8 or 1 mg OVB, depending on age and height, administered for 12 weeks. OVB was safe and effective in the treatment of EoE-EA. The current investigation focuses on the pharmacokinetics of budesonide and the impact of an oral viscous formulation on its absorption and bioavailability. Using a non-linear mixed effects approach, two distinct absorption profiles were identified, despite marked interindividual variability in drug concentrations. Budesonide exposure was higher than previously reported in children following oral inhalation. Even though no significant effect has been observed on serum cortisol levels, future studies should consider exploring different doses, schedules, and/or treatment durations, as there may be an opportunity to reduce the risk of cortisol suppression.

## 1. Introduction

Advancements in neonatal intensive care, anaesthesia, and surgical procedures have greatly improved the survival rates of individuals with oesophageal atresia (EA), an uncommon condition that impacts approximately 1 in 3500 live births in Europe [1,2]. Despite these advancements, individuals with EA are still prone to experiencing a range of complications, including gastro-oesophageal reflux disease (GERD), anastomotic stricture (AS), dysphagia accompanied by feeding disorders, and chronic respiratory diseases [3]. Moreover, in recent years, there has been an increasing focus on the association between EA and eosinophilic oesophagitis (EoE). The prevalence of EoE in individuals with EA is estimated to range from 9.5 to 30%, which is significantly higher than its prevalence in the general population of 0.1–0.5% [4,5,6,7,8,9]. Despite the ongoing debate about whether oesophageal eosinophilia in EA patients shares the same pathophysiological mechanisms as EoE in the general population [10,11], current evidence suggests that reducing mucosal eosinophil counts significantly improves dysphagia and GERD symptoms and lowers the occurrence of AS in EA children [6]. Even though fluticasone and oral viscous budesonide (OVB) are not approved for this indication in children, both products have been widely prescribed off-label, with an adequate efficacy and safety profile, i.e., inducing and maintaining clinical, endoscopic, and histological remission in patients with EoE for more than 15 years [12,13]. In view of the long-term nature of the treatment in these patients, one should question the dose rationale for this off-label indication and the potential implications of the systemic absorption of corticosteroids, including cortisol suppression and hypercortisolism.

Given its primary use in asthma, budesonide was developed to undergo high first-pass metabolism. This feature ensures that systemic exposure is limited and consequently less likely to induce adverse events. In fact, at the approved therapeutic dose range for asthma, budesonide shows an acceptable safety profile, despite its relatively short-lasting anti-inflammatory effect, as compared with other more potent corticosteroids, like for example, fluticasone [14,15]. Previously, Song and colleagues (2020) evaluated the systemic exposure to budesonide following administration of two budesonide oral formulations in adults [16], namely a budesonide oral suspension (BOS, 2 mg) and a gelatine capsule formulation (ENTOCORT EC, 9 mg). The study showed that systemic exposure to budesonide after a single oral dose varied not only with dose but also with dosage form. The area under the concentration vs. time curve (AUC∞) of the BOS 2 mg was less than the proportional dose relative to ENTOCORT EC 9 mg (geometric mean, 95% CI; 4.52, 3.61–5.67 vs. 13.74, 10.44–18.08 ng/mL·h) [16]. To date, the effect of age and body weight on the disposition of oral budesonide remains to be evaluated. Limited data are available in children and adolescents, for whom pharmacokinetics has been assessed only after the administration of budesonide oral suspension [17].

Recently, we conducted a phase 2 clinical trial in which the safety, pharmacokinetics, and efficacy of OVB in EoE-EA children was evaluated [18]. Regardless of the exploratory nature of the study, convincing evidence of treatment response, and overall safety profile over the 12-week period during which OVB was administered twice daily (Table 1), a major challenge has been the justification of the dose and dosing interval to be used in this clinical setting [19], which significantly differs from the currently approved indications for budesonide.

Here we use a model-based approach to characterize the pharmacokinetics of budesonide following administration of OVB in this small cohort of EoE-EA paediatric patients. In spite of the limited cohort size, the main objective of this investigation was to explore the influence of individual patient characteristics on the systemic concentrations of budesonide and assess the probability of adverse events, considering the known effect of the long-term use of corticosteroids in children [20].

## 2. Materials and Methods

### 2.1. Clinical Trial and Investigational Medicinal Product

A single centre, open-label, single-arm phase 2 clinical trial (EudraCT number 2019-002691-14) was previously conducted to assess the safety, pharmacokinetics and efficacy of OVB in *n* = 8 patients with EA and EoE [18]. The primary objective of this study was to evaluate the histological response to OVB therapy in patients with EoE-EA and to determine the proportion of patients achieving histological remission at study week 12 [18]. To minimise the sample size and control for futility, the study adopted an optimal Simon’s two-stage design, commonly used in phase 2 single-arm clinical trials. The study enrolled patients aged 3 to 18 years with primary EA repair, who underwent upper GI endoscopy (UGIE) following the current guidelines and were diagnosed with EoE based on international criteria [21,22]. 

At each study visit, safety assessments were performed, which included monitoring for any treatment-emergent adverse events (TEAEs) that occurred during the study period, physical examination, vital signs, body weight, height, and BMI, and laboratory investigations, including full blood counts and serum levels of creatinine, uric acid, transaminases, γ-GT, albumin, total and direct bilirubin, and C-reactive protein [18]. To test for adrenal suppression and hyperglycaemia, serum cortisol and glucose were measured at baseline, as well as at weeks 4, 8, and 12. 

Steady state sparse blood samples for the evaluation of budesonide concentrations were collected at week 12 (i.e., after 3 months of therapy) according to two sampling schedules, namely, (1) trough levels, immediately before the morning dose and at 30 min, 2 h, and 8 h after dose and (2) trough levels immediately before the morning dose and at 15 min, 1 h, and 4 h thereafter.

The study drug was a viscous formulation of budesonide supplied by I.T.C. FARMA S.r.l. (Pomezia, Italy), tailor-made for this study according to GMP requirements, with a viscosity of ≥2000 mPa∙s and a concentration of 0.2 mg/mL. The active pharmaceutical ingredient (API) was crystalline. Excipients included polysorbate, xylitol, sodium edetate, sodium citrate dihydrate, citric acid monohydrate, potassium sorbate, microcrystalline cellulose, sodium carboxymethyl cellulose, ascorbic acid, povidone, and water [18]. The product was kept at a temperature between 25 and 30 °C, protected from light, according to the manufacturer’s recommendations. Indeed, the closed product, while stored at temperatures ≤ 30 °C, has a shelf-life of 24 months. After opening of the glass bottles, which were provided with the pertaining expiration date, the product was used within 20 days and any remaining solution discarded, as prescribed by the manufacturer.

OVB was administered twice daily according to age, as previously reported [23,24,25,26,27]. Patients were stratified in three groups: 0.5 mg (3–4.9 years old), 0.8 mg (5–11.9 years old), and 1.0 mg (12–<18 years old). Patients were asked to avoid any food or water for at least 30 min following drug intake. Sparse blood samples were collected at different time points during the course of treatment, which lasted three months (12 weeks).

The study was approved by the Local Ethics Committee, and participants and/or their legal guardians provided written informed consent/assent before participating.

### 2.2. Bioanalytical Methods 

Budesonide quantitation was performed with a method previously used in a single centre, open-label, single-arm phase 2 clinical trial (EudraCT number 2019-002691-14) conducted to assess safety, pharmacokinetics, and efficacy of OVB in *n* = 8 patients with EA and EoE [18]. In particular, budesonide plasma samples were analysed using a liquid chromatography–mass spectrometry (LC-MS) system (i.e., an Agilent 1290 Infinity II UHPLC coupled with a 6470 Mass Spectrometer featuring an ESI-JET- STREAM source operating in positive ion mode (ESI+), (Agilent Technologies Deutschland GmbH, Waldbronn, Germany). Data analysis was performed using the MassHunter Workstation software 10.1 (Agilent Technologies). The linear calibration curve for budesonide ranged from 0.1 to 50 ng/mL, using budesonide-D8 as the internal reference standard (IS). The bioanalytical method was validated in accordance with EMA guidelines for bioanalytical methods validation [28]. For details on determining the plasma levels of budesonide, see the Appendix A.

### 2.3. Population Pharmacokinetic Modelling

The pharmacokinetic analysis of OVB was performed using a non-linear mixed effects modelling approach. Given the sparse nature of available samples and the small number of patients, a Bayesian approach was implemented in conjunction with a first-order conditional estimation with interaction (FOCE-I), which enables the use of historical data and the incorporation of priors into the estimation and minimisation procedures [29]. Considering the extensive pharmacological and biological knowledge on budesonide disposition, a structural model based on two-compartment disposition kinetics and two parallel absorption processes was selected to describe the disposition properties of budesonide [30]. The model was further refined considering the differences in formulation and population characteristics. First, the effect of informative, less-informative, and non-informative prior parameter distributions on the parameter estimates was tested. Second, the adequacy of parallel and/or sequential absorption models was considered (i.e., zero-order process combined with a first-order process, with or without a lag time). After identification of a suitable pharmacokinetic model, a sensitivity analysis was implemented to explore the impact of varying absorption processes and describe the impact of differences in the prevailing route of absorption. This step was based on random permutations and a variable contribution of zero- or first-order processes to the overall drug absorption. In addition, the effect of body weight and size differences on drug disposition parameters were assumed to vary according to allometric principles (i.e., power model based on body weight) [31].

Diagnostics and assessments of model performance were based on biological plausibility, a successful minimisation and covariance step, precision of the parameter estimates, standard goodness-of-fit plots (predicted versus observed, individual predicted versus observed concentrations, conditional weighted residuals over the range of population, and individual predicted concentrations), and visual predictive checks (VPCs). We then used the final model to generate post-hoc individual concentration vs. time profiles for each patient (*n* = 8). Secondary pharmacokinetic parameters (i.e., the area under the concentration curve from time 0 to 12 h (AUC_0–12_)) and maximum concentration (C_max_) were subsequently derived by non-compartmental methods. Model parameters were eventually used to simulate systemic exposure to budesonide, taking into account known thresholds for cortisol suppression and reduced growth velocity [32]. 

All data handling, graphical, and tabular summaries were performed in R (v.4.2.2) and R studio [33]. Modelling and simulation steps were implemented in NONMEM v.7.5 (ICON Development Solutions, Ellicott City, MD, USA), in combination with PsN-Toolkit (v. 5.3.0) and Pirana v.3.0.0. 

## 3. Results

### 3.1. Demographic Characteristics and Clinical Trial Findings

Baseline patient characteristics are summarised in Table 1. Five patients in the 5–11.9 years group received 0.8 mg of oral viscous budesonide every 12 h, and three patients in the 12–18 years group received 1 mg of oral viscous budesonide every 12 h. All patients, except one, reported regular intake of the study drug. After 12 weeks of treatment, all patients, except C-03, showed histological remission of EoE with a significant reduction in the median number of oesophageal eosinophils [18]. Throughout the study period, physical examination, vital signs, and laboratory workup (including cortisol values) remained within normal limits for all patients [18]. Glucose serum levels at the beginning (week 0) and the end of budesonide intake (week 12) were (median, IQR) 4.7 (4.6–5.1) mmol/L and 4.6 (4.5–5.0) mmol/L (*p* = 0.4909), respectively. Cortisol serum levels at baseline (week 0) were 309.1 (252.5–485.8) nmol/L, while they were 229.1 (220.0–391.9) nmol/L at study end (week 12, *p* = 0.5626). Detailed cortisol values for each patient have already been reported [18]. 

### 3.2. Pharmacokinetic Analysis

A total of 32 samples (4 per subject) were available. The observed concentration vs. time profiles revealed the presence of two distinct groups, with two distinct absorption processes and high interindividual variability (Figure 1). Thanks to the availability of a previously developed structural model describing the systemic disposition of budesonide, it was possible to explore and identify suitable parameterisation of the absorptive processes. Data were best described by a two-compartment model with parallel zero- and first-order absorption with a lag time (ALAG2) indicating a delay for the onset of the zero-order process, and first-order elimination (Figure 2). This parameterisation allowed us to estimate the fraction of first (F1)- and zero (F2)-order absorption process, the duration of the zero-order absorption (D2), and the absorption rate constant (Ka). Interindividual variability was retained for clearance (CL), central (V2), and peripheral (V3) volumes of distribution and intercompartmental clearance (Q).

CL, V2, Q, and V3 were allometrically scaled to account for the effect of body weight in the paediatric population. Model fit was improved by integrating non-informative priors on CL and V2 and corresponding interindividual variabilities, along with informative priors on Q and V3 and corresponding interindividual variabilities. A proportional error model was used to describe the unexplained residual variability.

Parameter estimates were found to be biologically plausible, with goodness-of-fit, VPCs, and other relevant statistical diagnostic criteria showing satisfactory performance of the model (Appendix A). In addition, the model allowed the characterisation of both population and individual predicted concentration profiles in all eight patients, whose data could be clustered into two distinct groups (Appendix A). Interestingly, the different profiles were not explained by the patient’s age, body weight, height, budesonide dose or sampling schedule. Rather, our analysis suggests that such patterns are associated with differences in absorption, with some subjects showing predominantly a zero-order process, whilst the first-order process prevails in others (Appendix A). 

The interquartile range of the secondary pharmacokinetic parameters show the magnitude of interindividual variability in the pharmacokinetics of budesonide, which was found to be dose independent (median and interquartile range, IQR). The predicted peak concentrations (C_max_) and area-under-the-concentration-time-curve (AUC_0–12_) varied by approximately 5-fold. Median C_max_ was 4.7 [0.9–4.9] ng/mL, whereas median AUC_0–12_ was 48.3 [8.2–51.9] ng/mL·h (Table 2).

## 4. Discussion

In EoE, steroid administration is aimed at topical effects on the oesophageal mucosa. The investigated viscous formulation is supposed to adhere to the oesophagus, maintaining the drug on the mucosa over a longer period of time, enhancing its ability to act on local inflammation, without the need for a systemic absorption or effect.

This investigation is the first attempt to characterise the pharmacokinetics of OVB in paediatric EoE-EA patients. Our analysis has provided insight into three main areas of interest, which should be considered for further evaluation of viscous formulations for EoE-EA patients. First, as observed in the individual data, the concentration versus time profiles reveal the presence of two distinct groups, pointing to the involvement of at least two different absorption processes. Second, interestingly, the simulations suggest that the prevalent absorption process may vary over the course of treatment, at different dosing events within the same individual, which may be explained by physical factors (e.g., standing, recumbent, or supine position) as well as variable peristaltic patterns, which may induce a slower or faster transit of particles from the oesophagus to the stomach. Third, despite the well tolerated profile during the course of 12 weeks, the systemic exposure to budesonide corresponded to approximately five times more than with inhaled budesonide administration in asthma [30].

The predicted exposure also corresponds to approximately 3 to 10 times higher AUCs as compared to the exposure observed following the administration of budesonide capsules to Crohn’s disease patients [34,35]. C_max_ and AUC_0–12_ values were between 5 and 20 times higher than those described for budesonide following oral inhalation in patients with asthma [36] or healthy subjects [37]. Intriguingly, both C_max_ and AUC were approximately 10 times higher than an oral suspension developed for EoE children [17]. This is a potentially undesirable occurrence, because systemic absorption, while not needed for efficacy, may foster side effects on steroid homeostasis and metabolism, including long-term effects on growth velocity, bone mineral density, immune system, blood pressure, and insulin sensitivity. Reassuringly, no patient developed the clinical manifestations of Cushing’s syndrome, diabetes mellitus, arterial hypertension or increased infectious diathesis. However, detailed cortisol values for each patient have already been published in a previous manuscript focusing on the efficacy and safety of the studied OVB formulation [18]. Nevertheless, we appreciate that the effects recorded over the 12-week study duration cannot be extrapolated with sufficient confidence over a much longer interval. We acknowledge that, ideally, a longer observation period and ACTH stimulation testing might be useful in future studies to corroborate these findings. Yet, plasma cortisol is a well-established marker of unwanted systemic steroid exposure for long-term inhaled corticosteroids [15,38] and there is no reason to assume that this should be different for viscous oral steroid formulations in EoE. Importantly, previous work has shown that growth inhibition is unlikely to occur without detectable reductions in plasma cortisol concentrations [32].

Based on physiological considerations, prior knowledge on budesonide disposition, and available pharmacokinetic models, we have identified two absorption processes, with a zero-order oesophageal absorption and a first-order gastro-intestinal absorption. The predicted profiles showed some individuals with a prevalent zero-order and others with a prevalent first-order absorption (Figure 1 and Appendix A). Interestingly, through simulations, we have also shown that inter-occasion variability may play a more relevant role than interindividual variability (Appendix A). In turn, this may explain why we are unable, with the limited number of samples and subjects available, to fully explain pharmacokinetic variability. Clearly, future studies on the pharmacokinetics of OVB should consider richer sampling schemes to ensure the characterisation of the underlying absorption processes. The physiological reasons for inter-occasion variability remain speculative, and could include varying posture, peristalsis, oesophageal motility, mucosal permeability, erosions, inflammation, and/or mucosal and submucosal blood flow.

Drugs absorbed through the gastro-intestinal tract are known to undergo an extensive first-pass effect, which in the case of budesonide leads to an oral bioavailability of approximately 10–15% [39]. As opposed to previous studies [17], in the current investigation, a fraction of the dose is absorbed by the oesophageal mucosa, which does not undergo a first-pass effect. Therefore, it is plausible that formulation and disease-related (i.e., tissue inflammation status) factors contribute to the higher systemic exposure.

The fairly long estimated duration of the zero-order absorption (approximately 8 h) points to a relevant permanence of the viscous formulation in the oesophagus. This mirrors what is known for budesonide oral capsules used for Crohn’s disease, which are retained on the intestinal mucosa [34]. Furthermore, given its action through nuclear receptors and translation, steroid-mediated inhibition of chemotaxis may outlast mucosal drug persistence. These two elements raise the question whether a once daily administration may lead to similar clinical effects. The absence of pharmacodynamic markers (apart from histologic remission at study end, achieved in all but one participant [18]), relatively short follow-up period, and the limited number of patients did not allow us to explore this opportunity, which will be worth addressing in future investigations.

Given the steady state concentrations observed in the current study, there is a significant risk of cortisol suppression and growth velocity reduction in those whose budesonide levels remain higher than 7.5 ng/mL. As systemic exposure does not vary linearly with the oral dose, the use of higher doses based on a patient’s height is not warranted. A 0.25 mg dose twice daily (or even 0.5 mg once daily) should be considered across the overall population, irrespective of age, height or body weight. In addition, it would be recommended to keep patients in the supine position for at least 15 min immediately after drug administration to minimise gastric absorption and consequently unnecessarily high systemic levels.

### Limitations and Strengths

Firstly, the narrow dose range used in this study has limited our ability to characterise linearity and non-linearity in drug disposition. Second, the concentration–response relationship could not be assessed. While this study showed that the selected doses were efficacious and well tolerated, it might be that lower doses yield a comparable effect. In this context, there may also be an opportunity for a once daily regimen, which would further simplify drug intake and enhance compliance, an issue of great relevance, particularly in children affected by chronic diseases like EoE [40].

We also acknowledge that the small sample size reduced the ability to identify other potential sources of variability. Similarly, the small sample size and the limited number of samples per patient forced us to rely more heavily on prior knowledge obtained in a different population. Therefore, the incorporation of prior knowledge is a suitable approach to deal with this limitation that is typical of a rare disease. Despite these limitations, the proposed model appears to describe the observed data satisfactorily and highlights the magnitude of interindividual differences in absorption and bioavailability. It also provides estimates of systemic exposure, which can be correlated with long-term effects, even though extrapolation of these results to a longer treatment period, beyond 12 weeks, cannot be validated with the current data.

Regardless of the aforementioned limitations, the use of a model-based approach allowed us to explore and characterise the absorption profile of this new oral viscous budesonide formulation and propose recommendations supporting the dose rationale for the treatment of paediatric patients with eosinophilic oesophagitis in repaired oesophageal atresia.

## 5. Conclusions

This study was the first to evaluate the pharmacokinetics of an oral viscous budesonide formulation in paediatric EoE-EA patients. Despite the small sample size, the observed patterns allowed the identification of different absorption processes, presumably linked to a combined oesophageal (zero-order) and intestinal (first-order) absorption. Interindividual and inter-occasion variability estimates were high and could not be explained by covariate factors known to affect drug disposition in children. Whilst a favourable efficacy and tolerability profile has been observed during the course of the 12-week period, our analysis shows that systemic exposure is considerably higher than previously demonstrated in paediatric asthma and Crohn’s disease patients.

Future studies are required to optimise the dosing regimen (i.e., once vs. twice daily, supine position post dose administration) and establish the dose rationale for OVB in paediatric EoE-EA patients. In addition, there is an opportunity to explore the implications of lower doses along with a more robust dosing algorithm using pharmacokinetic modelling and simulation principles.

## Figures and Tables

**Figure 1 pharmaceutics-16-00872-f001:**
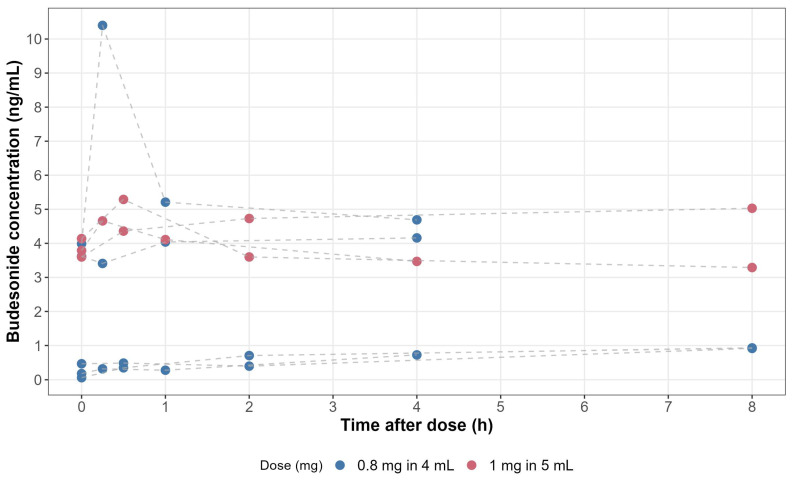
Observed budesonide plasma concentrations over time in patients treated with twice daily doses of 0.8 mg (blue dots) or 1.0 mg (red dots) oral viscous budesonide.

**Figure 2 pharmaceutics-16-00872-f002:**
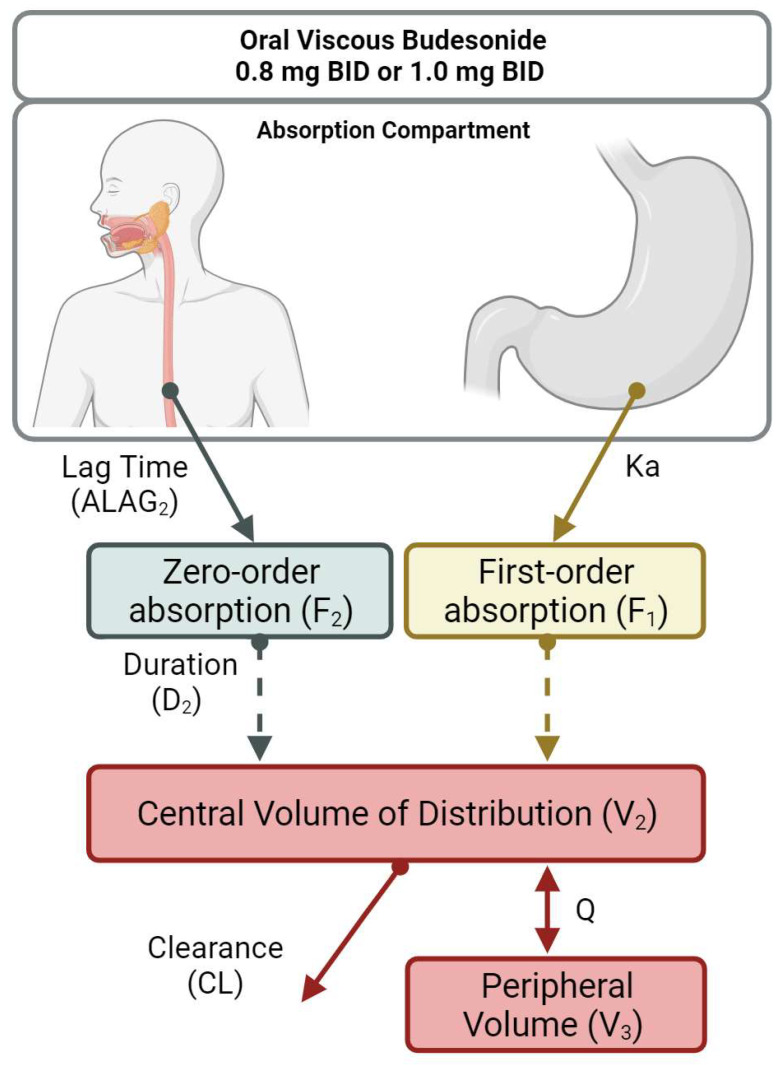
Schematic representation of the structural pharmacokinetic model. Parallel zero-order and first-order absorption processes with first-order elimination. Two different absorption processes take place following oral administration of budesonide. Initially, a fraction of the dose (F2) is absorbed in the oesophagus by zero-order kinetics, which progresses over a given period of time (D2), following a lag time (ALAG2). The remaining fraction of the dose (F1 = 1 − F2) is absorbed elsewhere in the gastrointestinal tract by first-order kinetics, characterised by the absorption rate constant Ka. Absorbed budesonide equilibrates in a central (V2) and a peripheral (V3) compartment according to an intercompartmental clearance (Q). Elimination from the central compartment (V2) occurs through a first-order process (CL).

**Table 1 pharmaceutics-16-00872-t001:** Baseline patient demographic and clinical characteristics.

Patient ID	Age at Baseline (Years)	Age at Diagnosis (Years)	Sex	Weight (kg)	Height (cm)	BMI (kg/m^2^)	BSA (m^2^)	Budesonide Schedule	Daily Budesonide Dose (mg/kg/d)
B-01	10.9	6.0	F	33.0	142	16.3	1.15	0.8 mg BID	0.05
B-02	6.2	3.1	F	20.0	118	14.3	0.82	0.8 mg BID	0.08
B-03	6.6	2.1	M	31.0	124	20.0	1.02	0.8 mg BID	0.05
B-04	7.3	3.2	F	23.5	129	14.2	0.93	0.8 mg BID	0.07
B-05	5.1	2.3	M	20.0	111.5	16.0	0.78	0.8 mg BID	0.08
C-01	12.0	11.8	F	55.9	149	25.0	1.49	1 mg BID	0.04
C-02	12.1	11.2	M	48.5	150	21.0	1.41	1 mg BID	0.04
C-03	16.9	14.2	F	53.0	155	22.0	1.50	1 mg BID	0.04
Median (Q1–Q3)	9.1 (6.5–12.0)	4.6 (2.9–11.3)	-	32 (22.6–49.6)	135.5 (122.5–149.3)	18.2 (15.6–21.3)	1.09 (0.88–1.45)	-	0.05 (0.04–0.075)

Abbreviations: AS, anastomotic stricture; BMI, body mass index; EA, oesophageal atresia; EoE, eosinophilic oesophagitis; PPI, proton-pump inhibitors.

**Table 2 pharmaceutics-16-00872-t002:** Secondary pharmacokinetic parameter estimates in EoE-EA patients (*n* = 8).

Participant	Dose [mg]	C_max_ [ng/mL]	AUC_0–12_ [ng·h/mL]
1	0.8	0.9	8.3
2	0.8	0.9	7.7
3	0.8	0.8	6.7
4	0.8	4.9	51.1
5	0.8	8.6	62.3
6	1.0	4.8	49.5
7	1.0	4.5	47.0
8	1.0	5.0	54.4
Median (interquartile range)	-	4.7 (0.9–4.9)	48.3 (8.2–51.9)

## Data Availability

Data will be available upon request to corresponding authors.

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
