# Peer review of "Pharmacokinetic Evaluation of Oral Viscous Budesonide in Paediatric Patients with Eosinophilic Oesophagitis in Repaired Oesophageal Atresia"

_pharmaceutics, 2024, doi:10.3390/pharmaceutics16070872_

Round 1
Reviewer 1 Report
Comments and Suggestions for Authors
The merit of the work is high, and the topic is very interesting. Esophageal-retentive budesonide formulations for EoE treatment in children must be understood better. The pharmacokinetic data on this subject is rare, and the popPK models are even rarer. Therefore, the work is original.
Major issues:- more information is required on the OBV formulation - was it a solution or suspension, what excipients were used, or how was it compounded, what was the viscosity, the API type (amorphous or crystalline?). If such data are available, please include them in the manuscript.
- have the authors tried using a transit-compartment model (Erlang-type)?
- the sampling schedule should be described better. For example, the authors do not state clearly that the samples were taken in steady-state (Day X?) on a single day.
- The results provided by the authors are very intriguing in light of the available literature data. A product with a similar indication was recently approved by the FDA (https://www.accessdata.fda.gov/drugsatfda_docs/label/2024/213976s000lbl.pdf). It is also a viscous product, suspension type (this is why I require more information on the formulation tested by the authors). The label includes data from pediatric subjects (aged 11 years to 17 years). The PK results after oral administration of EOHILIA 2 mg twice a day were: the median (range) time to peak plasma concentration of budesonide was 1 (0.5, 2) hour, the mean (%CV) Cmax was 946 (61%) pg/mL and the mean (%CV) AUC0-8h was 3849 (51%) pg•h/mL. They align well with the first three subjects of the current study. It is quite confusing to see such high exposures in the rest of the children! Where could these differences come from? EOHILIA trial also included children with EoE, so mucosal permeability, erosions, inflammation and/or mucosal and submucosal blood flow could be similar. Was the body position (standing, supine, seated) monitored in those children with high exposures? I recommend double-checking the underlying concentration data and going back to the chromatographic analysis. Perhaps inspecting individual chromatograms, calculations or even reanalyzing incurred samples could explain it.
- Figure 1 and the popPK diagnostic plots show that one patient had C> 10 ng/mL. However, Table 2 shows that the maximum Cmax was 8.6 ng/mL. Another subject (one of the 1 mg dose group) had Cmax around 5.2 ng/mL, according to Figure 1. However, the Cmax shown in Table 2 is only 5.0 ng/mL. The VPC graph shows yet another set of observed data points, exceeding 15 ng/mL! It makes me doubt the data used in this manuscript and the quality of the analysis.
- line 240: this statement: "Second, interestingly, the data..." cannot be drawn from the data. The authors did not analyze budesonide concentrations over several occasions in a single subject. Although it is possible and even highly probable, it is not what the current study shows.
- line 244: please be cautious when placing such a statement. These were not only different routes of administration but also different doses. If any comparisons were to be made, the concentrations should be dose-adjusted (assuming PK linearity).
Minor issues:
- line 38: "or" instead of "org"
- line 70: the data is incorrect, it should state "5.67" instead of "5.66"
- Table 1: please specify, that the last column shows daily budesonide dose
- lines 184 & 189: references in these lines were not found by the system, please correct it
Author Response
Comments and Suggestions for Authors
The merit of the work is high, and the topic is very interesting. Esophageal-retentive budesonide formulations for EoE treatment in children must be understood better. The pharmacokinetic data on this subject is rare, and the popPK models are even rarer. Therefore, the work is original.
# Many thanks for your positive comments.
- Major issues:- more information is required on the OBV formulation - was it a solution or suspension, what excipients were used, or how was it compounded, what was the viscosity, the API type (amorphous or crystalline?). If such data are available, please include them in the manuscript.
# Thank you for your question. Details on the OVB formulation used in our study were reported in a previous publication [Tambucci R et al., J Pediatr Gastroenterol Nutr. 2023;77(2):249-255. doi:10.1097/MPG.0000000000003830]. Briefly, the budesonide formulation used in this study was a suspension with a viscosity of ≥ 2000 mPa∙s and a concentration of 0.2 mg/mL. The active pharmaceutical ingredient was crystalline. Excipients included: polysorbate, xylitol, sodium edetate, sodium citrate dihydrate, citric acid monohydrate, potassium sorbate, microcrystalline cellulose, sodium carboxymethyl cellulose, ascorbic acid, povidone, and water. The final product was stored at a temperature of +25 to +30 °C, protected from light. This information has been included in the revised versin of the manuscript (section Methods).
- Have the authors tried using a transit-compartment model (Erlang-type)?
# Thank you for this pertaining comment. Yes, we did. However, this was problematic. Notably, we disposed about a very limited set of data. The implementation of an Erlang type (manually coded number of transit compartments) absorption or of a transit compartment model (flexible number of transit compartments), which we tried, resulted difficult (or required to fix several parameters), actually unsurprisingly, because such a structure requires a thorough sampling and an intensive stochastic parameterisation (without adding, however, a mechanistic explanation for the underlying biological factors). The available dataset (8 patients, 4 samples per patient) did not allow to succesfully complete this task. Rather, we needed to rely on a previously published and validated model, exploiting the hereby identified model structure and making use of priors. In addition, the selected parameterisation allowed us to better interpret the observed interindividual differences.
- The sampling schedule should be described better. For example, the authors do not state clearly that the samples were taken in steady-state (Day X?) on a single day.
# Thank you for your pertaining comment. We apologise for not having stated this clearly in the initial manuscript. According to the study protocol (EudraCT number: 2019-002691-14), PK sampling was performed at steady state, after 3 months of therapy (study week 12), with two different sampling schedules. This is now explitely stated in the revised manuscript (section Methods).
- The results provided by the authors are very intriguing in light of the available literature data. A product with a similar indication was recently approved by the FDA (https://www.accessdata.fda.gov/drugsatfda_docs/label/2024/213976s000lbl.pdf). It is also a viscous product, suspension type (this is why I require more information on the formulation tested by the authors). The label includes data from pediatric subjects (aged 11 years to 17 years). The PK results after oral administration of EOHILIA 2 mg twice a day were: the median (range) time to peak plasma concentration of budesonide was 1 (0.5, 2) hour, the mean (%CV) Cmax was 946 (61%) pg/mL and the mean (%CV) AUC0-8h was 3849 (51%) pg•h/mL. They align well with the first three subjects of the current study. It is quite confusing to see such high exposures in the rest of the children! Where could these differences come from? EOHILIA trial also included children with EoE, so mucosal permeability, erosions, inflammation and/or mucosal and submucosal blood flow could be similar. Was the body position (standing, supine, seated) monitored in those children with high exposures? I recommend double-checking the underlying concentration data and going back to the chromatographic analysis. Perhaps inspecting individual chromatograms, calculations or even reanalyzing incurred samples could explain it.
# Thank you for this interesting question. Actually, we were surprised about the measured concentrations as well! However, we again double-checked the LC-MS/MS results with the laboratory, and the pre-analytical process with the clinical study team, and we did not detect any laboratory error, nor we identified any pre-analytical bias source. It is difficult to identify with certainty the reason (or the multiple reasons) for the differences between the studied formulation and EOHILIA.
We hypothesize that this may be due to formulation characteristics, although we cannot identify them specifically. However, we concur with the potential impact of gravity, as viscosity may not be sufficient to retain the dose on the esophageal mucosa if subjects are standing. This point has been brought up in the discussion. The participants of our study were asked not to drink and not to eat during the 30min following the drug intake, but were not requested to remain supine or seated (i.e. they were allowed to follow their daily activities). These details are highlighted in the revised manuscript.
- Figure 1 and the popPK diagnostic plots show that one patient had C> 10 ng/mL. However, Table 2 shows that the maximum Cmax was 8.6 ng/mL. Another subject (one of the 1 mg dose group) had Cmax around 5.2 ng/mL, according to Figure 1. However, the Cmax shown in Table 2 is only 5.0 ng/mL. The VPC graph shows yet another set of observed data points, exceeding 15 ng/mL! It makes me doubt the data used in this manuscript and the quality of the analysis.
# Thank you for your questions. Table 2, as its title states, reports “Secondary pharmacokinetic parameter estimates”: these are estimates based on the final model (Individual predicted concentration vs. time profiles as shown in Supplementary Figure S3), while Figure 1 reports the raw measured data. We hope this helps explaining the apparent inconsistency between Figure 1 and Table 2.
An old, incorrect version of the Supplementary Figure 2 (VPC) was unfortunately uploaded. We are very sorry about this, and we warmly thank the reviewer for having picked up this mistake. The correct figure has now been uploaded (Supplementary Figure S2), which is consistent with Figure 1.
- line 240: this statement: "Second, interestingly, the data..." cannot be drawn from the data. The authors did not analyze budesonide concentrations over several occasions in a single subject. Although it is possible and even highly probable, it is not what the current study shows.
# The Reviewer is right: we did not measure budesonide concentrations over several occasions. However, basing on the finally developed model, we simulated the behavior of budesonide and the obtained concentrations, in several occasions, simulating different absorption patterns (prevalently first- or prevalently zero-order absorption) over the preceding 5 dose intakes, as depicted in Supplementary Figure S3. It is on these simulation scenarios that we base(d) our conclusions. In the revised manuscript (Discussion), this is now more transparently stated.
- line 244: please be cautious when placing such a statement. These were not only different routes of administration but also different doses. If any comparisons were to be made, the concentrations should be dose-adjusted (assuming PK linearity).
#Thank you for this important comment. We fully agree. Indeed, we are sorry that we did not already address this point in the original submission, and we are very grateful to the reviewer for bringing our attention to this point. In the original publication by Soulele and colleagues (reference #30), participants inhaled 400?g of budesonide, which provided a Cmax of 725.35pg/mL. If we compare this with the 0.8mg (1450.71 pg/mL = 1.45 ng/mL) or 1mg (1813.39pg/mL = 1.81 ng/mL) our participants swallowed, the ratio is of approximately 5 times higher. In the revised manuscript, we corrected the statement accordingly (“approximately 5 times” instead of “approximately 10 times”).
Minor issues:
- line 38: "or" instead of "org"
# Done. Thank you for your careful eyes!
- line 70: the data is incorrect, it should state "5.67" instead of "5.66"
# Done. Thank you again!
- Table 1: please specify, that the last column shows daily budesonide dose
# Done. Thank you for this constructive comment.
- lines 184 & 189: references in these lines were not found by the system, please correct it
We do apologies for this misleading point. In the revised version:
- old version: “high interindividual variability (Error! Reference source not found.).”
- corrected version (Line 208): “high interindividual variability (Figure 1).”
- old version: “and first-order elimination (Error! Refer-189 ence source not found.).”
- corrected version (Line 213): “and first-order elimination (Figure 2).”
Reviewer 2 Report
Comments and Suggestions for Authors
This is an interesting and difficult to realize study where the authors characterize the pharmacokinetics of oral budesonide in pediatric population. Indeed, pharmacokinetic studies are difficult to be realized in children, which explains the small number of included subjects and the high interindividual variability.
Minor:
- lines 184, 189: (Error! Reference source not found.)
Author Response
- Lines 184, 189: (Error! Reference source not found.)
Thank you for your careful eyes. In the revised manuscript, this has been corrected.
Reviewer 3 Report
Comments and Suggestions for Authors
In the current article, the authors conducted an exploratory study concerning paediatric eosinophilic esophagitis after esophageal atresia to assess the safety, pharmacokinetics and efficacy of oral viscous budesonide, twice-daily administered.
Some suggestions:
1. Abstract – Please add the duration of the treatment with oral viscous budesonide.
2. pg 1, line 40-41- you wrote that esophageal atresia it is: ”an uncommon condition that impacts approximately 1 in 3,000 infants”. Is this incidence worldwide? Please clarify.
3. pg 2, line 62-63 - you wrote that ”budesonide shows acceptable safety profile, despite its relatively short-lasting anti-inflammatory effect, as compared with other more potent corticosteroids”. Add please which are these corticosteroids.4. pg 2, line 79: Give please details concerning the dose and dosing interval.
5. pg 3 lines 103-104 - you wrote that ”To test for adrenal suppression and hyperglycaemia, serum cortisol and glucose levels were repeatedly measured throughout the study”. Please add: - the time intervals at which serum glucose and cortisol levels were determined (you wrote different time)
-the names of the devices used to make these determinations and their origin
- the methods by which the determinations were performed.
Did you determined also other biochemical parameters? 6. pg 3, lines 111-112 - you wrote that ” The product was kept at a temperature between 25 and 30 °C, protected from light”. How long did you keep the product at a temperature between 25 and 30°C. What do you know about the stability of the product? Please clarify. 7. pg 3, lines 113-114- How did you correctly establish the dose of administered budesonide?You wrote: “Patients were stratified in three groups: a 0.5 mg (3-4.9 years old), 0.8 mg (5-11.9 years 114 old) and 1.0 mg (12-<18 years old)”. In my opinion is not correct to establish the amount of budesonid administered only according to the patient age you must also take into account the patient's weight.
8. Figures 1 and 2 are not presented at pages 5 and 6.
9. From the article are missing the obtained values for blood glucose and cortisol. Please add them and also discuss them.
A great limitation of your study is the fact that is carried out on a small number of patients. Also the number of samples from which budesonide was determined is small (4 per subject).
Comments on the Quality of English Language
Minor editing of English language is required.
Author Response
Abstract – Please add the duration of the treatment with oral viscous budesonide.
# Thank you for your suggestion. This has been integrated in the revised manuscript.
pg 1, line 40-41- you wrote that esophageal atresia it is: ”an uncommon condition that impacts approximately 1 in 3,000 infants”. Is this incidence worldwide? Please clarify.
# Thank you for your comment. In the revised manuscript, we now state: “approximately 1 in 3,500 live births in Europe”. Furthermore, we added following reference:
[Sfeir R, Aumar M, Sharma D, Labreuche J, Dauchet L, Gottrand F. The French Experience with a Population-Based Esophageal Atresia Registry (RENATO). Eur J Pediatr Surg. 2024 Apr;34(2):137-142. doi: 10.1055/a-2206-6837. Epub 2023 Nov 8. PMID: 37940126].
pg 2, line 62-63 - you wrote that ”budesonide shows acceptable safety profile, despite its relatively short-lasting anti-inflammatory effect, as compared with other more potent corticosteroids”. Add please which are these corticosteroids.
# Thank you for your comment. In the revised manuscript, we now cite an example of such corticosteroids (i.e fluticasone propionate).
pg 2, line 79: Give please details concerning the dose and dosing interval.
# Thank you for this comment. We now reference, also at this point of the manuscript, to Table 1, which details the administered doses. The dosing interval (“twice-daily”) is provided in the preceding line (line 80 of the revised manuscript).
pg 3 lines 103-104 - you wrote that ”To test for adrenal suppression and hyperglycaemia, serum cortisol and glucose levels were repeatedly measured throughout the study”. Please add:
# Thank you for this pertinent question. In the revised mansucript, we now provide this information:
- the time intervals at which serum glucose and cortisol levels were determined (you wrote different time)
# Both cortisol and glucose levels were evaluated at baseline (week 0) and at weeks 4, 8, 12 (Lines 108-109). In the revised manuscript, we now report glucose and cortisol values measured at baseline (week 0) and study end (week 12), in order to show the absence of hyperglycaemia and cortisol suppression (Paragraph 3.1. Demographic characteristics and clinical trial findings of Results section). Detailed cortisol values have already been published [Tambucci R et al., J Pediatr Gastroenterol Nutr. 2023;77(2):249-255. doi:10.1097/MPG.0000000000003830].
- the names of the devices used to make these determinations and their origin
# Cortisol, glucose and other biochemical parameters were measured at each study visit, by using an automated laboratory platform named “The Integrated Core Laboratory” that includes Cobas® c 702 and 801 instruments (Roche Diagnostics GmbH, Engelhorngasse 3, Vienna, Austria). In the revised manuscript, the pertaining details have now been included (Materials and Methods section).
- the methods by which the determinations were performed.
# Cobas® system is a fully automated immunochemical platform.
- Did you determined also other biochemical parameters?
# We did determine also further biochemical parameters. Indeed, each study visit also included a full blood count and the determination of creatinine, urea, liver transaminases, γ-GT, albumin, total and direct bilirubin, and C-reactive protein. However, these results have been previously reported. In this manuscript focusing on PK (and on drug effects on glucose), we would prefer not to repeat their publication here. If the Reviewer feels that they are key to this manuscript as well, we might ask the pubslisher of the efficacy and safety article [Tambucci R et al., J Pediatr Gastroenterol Nutr. 2023;77:249-255.] for permission to represent them also here.
pg 3, lines 111-112 - you wrote that ” The product was kept at a temperature between 25 and 30 °C, protected from light”. How long did you keep the product at a temperature between 25 and 30°C. What do you know about the stability of the product? Please clarify.
# Thank you for your pertinent question. According to the manufacturer’s instructions, the closed product, while stored at temperatures ≤30°C, has a shelf-life of 24 months. After opening of the glass bottles, which were provided with the pertaining expiration date, the product was used within 20 days, as prescribed by the manufacturer. This information has been added to the revised manuscript.
pg 3, lines 113-114- How did you correctly establish the dose of administered budesonide? You wrote: “Patients were stratified in three groups: a 0.5 mg (3-4.9 years old), 0.8 mg (5-11.9 years 114 old) and 1.0 mg (12-<18 years old)”. In my opinion is not correct to establish the amount of budesonid administered only according to the patient age you must also take into account the patient's weight.
# Thank you. We agree. Indeed, the aim of this study was to characterize the PK of the novel oral viscous budesonide formulation, so to be able to provide a more appropriate dose rationale. In fact, the chosen model included weight as a covariate describing oral viscous budesonide PK (Supplementary Table S1). However, in the Methods, we report how the administered dose was chosen, as per study protocol.
Figures 1 and 2 are not presented at pages 5 and 6.
#We are sorry, we do not fully understand this comment. In the pdf file we could download from the MDPI web-based system (“Susy”), Figure 1 appears on page 5 and Figure 2 on page 6 (now pages 6 and 7 in the revised manuscript).
From the article are missing the obtained values for blood glucose and cortisol. Please add them and also discuss them.
#Thank you for this question. Please also see answer to your comment #5. In the revised manuscript, we now report glucose and cortisol values measured at baseline (week 0) and at study end (week 12), which show the absence of hyperglycaemia and of cortisol suppression after 12 weeks of budesonide intake (Paragraph 3.1. Demographic characteristics and clinical trial findings of Results section, lines 199-203).
Detailed cortisol values for each patient have already been published in a previous manuscript focusing on efficacy and safety of the studied OVB formulation [Tambucci R et al., J Pediatr Gastroenterol Nutr. 2023;77(2):249-255. doi:10.1097/MPG.0000000000003830].
A great limitation of your study is the fact that is carried out on a small number of patients. Also the number of samples from which budesonide was determined is small (4 per subject).
# Thank you, we fully agree, but this is a rare condition. There is limited opportunity to explore large sample sizes. Actually, already in the original submission we tried to acknowledge this fact, and the consequent need of relying on a previously published PK model and making use of priors. In the revised manuscript, we now state this limitation more clearly, stressing not only the small participant number but also the very sparse sampling available.
Reviewer 4 Report
Comments and Suggestions for Authors
This is an interesting manuscript on the pharmacokinetic evaluation of budesonide in children with a history of repaaired esophageal atresia, that developed EoE.
Some minor suggestions;
Line 26, change "org" to "or"
Line 184 and 189, an action has to be made with the references
Line 231, use the EoE abbreviation
Line 253; you mention that both Cmax and AUC were higher thatn an oral suspension used by Gupta SK, et al. Do you mean that they had used another formulation of budesonide, or another "brand" of viscous product? You are mentioning that you have used a product supplied by I.T.C. FARMA. Is it a commercially available product, or a tailor-made for your study? Please provide information at Line 111.
Author Response
- Line 26, change "org" to "or"
Thank you for your careful eyes. This has been integrated in the revised manuscript.
- Line 184 and 189, an action has to be made with the references
Thank you. This has been corrected in the revised manuscript.
- Line 231, use the EoE abbreviation
Done. Now it becomes line 254.
- Line 253; you mention that both Cmax and AUC were higher than an oral suspension used by Gupta SK, et al. Do you mean that they had used another formulation of budesonide, or another "brand" of viscous product? You are mentioning that you have used a product supplied by I.T.C. FARMA. Is it a commercially available product, or a tailor-made for your study? Please provide information at Line 111.
# The viscous suspension used in the current study is a novel formulation, currently under development and not yet available as commercial product (this is now explicitely stated in the revised manuscript). It was described in a previous publication [Tambucci R et al., J Pediatr Gastroenterol Nutr. 2023;77(2):249-255], and its physical and chemical properties are now explicitely reported in the revised mansuscript (owing to a specific request of Reviewer #1).
The article by Gupta SK and colleagues does not specify neither the producer not the exact composition of the used formulation. However, the authors report that the “Budesonide oral suspension (BOS) is a swallowed, viscous, immediate-release topical corticosteroid developed for EoE and optimized to maximize mucosal contact at the esophageal surface.” In a previous publication by the same group, the authors specify that this was “a proprietary formulation containing budesonide”, without further details.
Round 2
Reviewer 1 Report
Comments and Suggestions for Authors
I have no further comments.